# Transcripts of Unfulfillment: A Study of Sexual Dysfunction and Dissatisfaction among Malay-Muslim Women in Malaysia

**Rosediani Muhamad** [1,*], **Dell Horey** [2], **Pranee Liamputtong** [3], **Wah Yun Low** [4], **Maryam Mohd Zulkifli** [1] and **Hatta Sidi** [5]

1   Family Medicine Department, School of Medical Sciences, Health Campus, Universiti Sains Malaysia, Kubang Kerian 16150, Malaysia; maryammz@usm.my
2   Department of Public Health, College of Science, Health and Engineering, Melbourne Campus, La Trobe University, Bundoora, VIC 3086, Australia; d.horey@latrobe.edu.au
3   Translational Health Research Institute (THRI), Western Sydney University, Campbelltown, Locked Bag 1797, Penrith, NSW 2751, Australia; P.Liamputtong@westernsydney.edu.au
4   Faculty of Medicine and Asia-Europe Institute, University of Malaya, Kuala Lumpur 50603, Malaysia; lowwy@um.edu.my
5   Department of Psychiatry, Pusat Perubatan University Kebangsaan Malaysia, Bangi 43600, Malaysia; hatta@hotmail.com
*   Correspondence: rosesyam@usm.my or drrosediani@gmail.com

**Abstract:** The prevalence of female sexual dysfunction (FSD), or in everyday notion, sexual dissatisfaction, among Malay women remains high, denoting that there are several influences shaping their experience of sex within marriage. This qualitative study identified the perceived effects of social factors in the development of sexual dysfunction among Malay women. Engaging a phenomenological framework, 26 in-depth face-to-face interviews were conducted among married women from Peninsular Malaysia, based on their self-reporting of FSD symptoms. All sessions were audio-recorded and the data were transcribed verbatim and managed in the ATLAS.ti software before being analysed. The three themes that emerged—'sex is taboo and culturally unacceptable', 'self-ignorance about sex', and 'lack of husband's role in mutual sexual enjoyment'—suggest some influence of Islamic teachings and cultural conduct, as in Adat, on sexuality in society. However, a lack of knowledge and nonadherence to positive values and teachings around sexual satisfaction between men and women, as espoused through the Islamic religion, have affected woman's sexual functions and coupling relationship even more significantly. The results of this qualitative study show that a formal, culturally sensitive, and comprehensive sex education programme incorporating both medical and Islamic knowledge may work to effectively reduce FSD.

**Keywords:** Islam; Malay women; female sexual dysfunction; sexual dissatisfaction; sexual relationship

## 1. Introduction

Malays have a unique legal identity formed by their religion (Islam), language (Bahasa Melayu), and customs (Adat), formalized in Malaysia's independence in 1957 (Federal Constitution 2010). These three components were brought together as a political solution in a geographical area with a complex history, but also contribute to shaping the sexual identity of Malay women. While Islam steers Malaysia through its beliefs, religious practices, and political responses, Adat underpins how people live and language holds everything together (Mastor et al. 2000). The process of integration, particularly of Islam and Adat, presents opportunities to influence change.

Despite its position as the official religion of Malaysia, Islam has not totally overruled Malay Adat. While mostly Adat and Islam work together, some Adat practices are non-sharia, that is, they do not comply with Islam (Harun 2009). However, over time non-sharia practices are being slowly modified to follow Islamic law, both through political

will and through the influence of Islam on what constitutes desirable personal attributes (Harun 2009). For example, Malays are associated with politeness and humility, which are viewed as positive characteristics and are actively reinforced by Islamic teaching. Government members and Islam scholars have initiated various ongoing campaigns to encourage values consistent with Islamic teaching to shape a respectful Malay society with high morality and a strong identity (Ahmad and Mohamad Nasir 2018; Malaysiakini 2018).

For Malay women, education of their sexual identity starts in girlhood through their family upbringing and local religious teachers and it continues through marriage and motherhood (Omar 1994). Apart from politeness and humility, loyalty is also seen as a trademark attribute of the Malay identity (Goddard 2000). Nevertheless, beyond these cultural attributes, limited knowledge of sex or ways to address deficiencies in sexual relationships, coupled with nonparticipation of husbands in resolving these issues, seem to underlie the problems experienced by Muslim women (Khoei et al. 2008; Muhamad et al. 2016).

Government programs that provide avenues to address these issues have been in place in Malaysia since 1996. Adolescent health services were started by the Ministry of Health in 1996, mainly within school health units, but these were not fully utilized until 2018 when the Ministry rebranded and strengthened them as adolescent-friendly health services in health clinics (Awang et al. 2020; Nik Farid et al. 2018). Since 2011, trained teachers have delivered the Health Reproduction and Social Education (PEERS) program in schools during Islamic studies, moral studies, and science and health education classes (Awang et al. 2020; Nik Farid et al. 2018). However, PEERS is not a comprehensive sexual health program and is still considered controversial among many parties (Mohd Mutalip and Mohamed 2012). The compulsory premarital courses, also introduced in 1996 by the Department of Islamic Development Malaysia, using an integrated module for premarital course, mainly covers marital roles and rights (Saidon et al. 2016).

Sexuality is complex and a broad topic. It refers to the way people feel and behave to express themselves as sexual beings (Ferrante 2014). Sexuality involves biological (genetic), erotic, and emotional (psychological) factors, including personal physique, social conditioning, and behavioural and spiritual influences (Bolin and Whelehan 2009). In Malaysia, sexuality is shaped from childhood as preparation for marriage and with consideration of the traditional adult Malay woman who needs to take account of her role as a wife. In Islam, sexual relationships between a man and a woman are only permitted within the nikah (marriage) (Khoei et al. 2008).

Once married, the status of women is higher and even valued, as reflected in many surahs (chapters) in the Quran, including An-Nisa' and An-Nur (Hamdan and Md. Radzi 2014). For example, the obligation to be a good partner is mutual for both men and women, but the rights and responsibilities of husbands and wives are different. The interplay of the context-based identity of Malays and the modernisation of Malaysia offers different possibilities to how Malay women may construct the meanings of sexuality and how they could react to sexual dysfunction, but there are no actual studies to show which of these possibilities actually occur.

While the Malay identity includes values that many consider positive, Malay women are found to be at higher risk of experiencing sexual dysfunction (Sidi et al. 2007). Female sexual dysfunction (FSD) refers to "a group of disorders that are characterised by a clinically significant disturbance in a person's ability to respond sexually or to experience sexual pleasure" (American Psychiatric Association (APA) 2013), which includes 'Female Orgasmic Disorder' (FOD), 'Female Sexual Interest/Arousal Disorder' (FSIAD), or 'Genito-Pelvic Pain/Penetration Disorder' (GPPD). The prevalence of FSD ranges from 25.8% to 30.0% among the general population in Malaysia (Ishak et al. 2010; Sidi et al. 2007) and goes up to 90% among breast cancer patients (Shuib 2014). This paper aimed to evaluate the causes behind sexual dysfunctionality or sexual dissatisfaction among Malay women.

## 2. Materials and Methods

*Study Design and Methods*

This article was based on a qualitative study that utilised a phenomenological framework. The study involved 26 Malay women who self-reported at least one of the following sexual dysfunction symptoms: (1) a lack or absence of sexual desire, (2) a lack or absence of sexual arousal, (3) trouble with or having no orgasm, or (4) sexual pain during intercourse. The women were purposefully recruited from Peninsular Malaysia from late 2013 to the middle of 2014. Most responded to flyers that had been posted online or distributed at health clinics under the Ministry of Health and various universities. Other recruiting strategies included doctor referrals, word-of-mouth through local leaders, and snowball sampling.

Those selected were heterosexual women above 18 years of age who had lived together with their husband for at least one year. Initially, 32 women agreed to join the study. However, six were eventually excluded due to other conditions, such as infertility, rather than only experiencing sexual dysfunction; refusing to be interviewed directly or face-to-face; or being unable to commit to the interview schedule.

After obtaining consent of the prospective interviewees, data were primarily collected through face-to-face in-depth interviews conducted by the first author. Among the questions asked were what these women perceived as reasons for their sexual dysfunction and the factors behind these. The interviews mainly took place in hospitals, while a few were done at homes or in restaurants. They were carried out by a Malay doctor using the Malay language to facilitate communication and ensure that both parties understood each other clearly. All interviews were audio-recorded and took around 60 min to be completed. Additional data included field notes written up right after the completion of each interview.

All interview audio-recordings were transcribed verbatim. To maintain confidentiality, participants were assigned pseudonyms and their respective workplaces anonymised. The transcripts were then transferred into the ATLAS.ti® programme for coding and identification of possible themes. Preliminary codes were generated from the literature in accordance with the research questions. The analysis began with the process of repeated readings of each transcript until the authors could make sense of each word and sentence. This process also helped us to recognise repeated patterns of meanings, which later could be used to make sense of their narratives (Fereday and Muir-Cochrane 2006; Liamputtong 2020). New codes were also created during these readings. The level and relationship of codes were identified. The axial coding was then created. Different codes were combined and rearranged to create suitable themes within and across participants' stories (Liamputtong 2020).

The researchers, two of whom are experts in qualitative research, maintained the validity of this study via thorough discussion and evaluation of the themes identified. Only quotes chosen to be used in the thesis were translated into English. Furthermore, these quotes, along with the transcripts, were emailed to participants for clarification and confirmation of the meanings understood. All agreed with the findings and some encouraged us to share these with the larger community.

## 3. Malay Women and 'Sexual Dysfunction'

*3.1. Background*

The profile of the 26 Malay women involved in this study is reported in Table 1 as well as in Muhamad et al. (2019). On average they were 39.5 years-old, had been married for almost 14 years, and the majority worked in public service. Close to 60 percent claimed to either have had problems with sexual desire or a mix of problems around sexual arousal, desire, and orgasms.

**Table 1.** Profile of participants (*n* = 26).

| Variable | Mean (SD) | N (%) |
|---|---|---|
| **Age** | 39.5 (7.7) | - |
| **Marriage duration in years** | 13.7 (6.3) | - |
| **Number of children** | 3.4 (6.3) | |
| **Education levels** | | |
| Tertiary | | 16 (61) |
| Below | | 10 (39) |
| **Occupation** | | |
| Working (government/private sectors) | | 21 (66) |
| Housewife/self-employed | | 5 (16) |
| **Types of sexual dysfunction** | | |
| Lack or loss of sexual desire | | 7 (26.9) |
| Difficulties with sexual arousal | | 0 (0) |
| Difficulties with orgasms | | 1 (3.9) |
| Pain during sex | | 3 (11.5) |
| Any of the above | | 15 (57.7) |

*3.2. Contributing Factors*

Broadly, the reasons the women gave for their experience with sexual dysfunction or sexual dissatisfaction could be categorised under three themes: (1) the belief that sex is taboo and culturally unacceptable, (2) self-ignorance about sex, and (3) lack of Muslim husband's role in mutual sexual enjoyment. The themes are in the Table 2.

**Table 2.** Themes generated according to influence of culture and religion on the aetiology of female sexual dysfunction (FSD) (*n* = 26).

| Themes | Sub-Themes | Descriptions |
|---|---|---|
| Sex is taboo and culturally unacceptable | Sex talk is disgraceful<br>Sex is dirty<br>Romantic love not widely practiced | • Sex is something taboo and less important<br>• Learned little about sexuality from their families<br>• Use of sex-related words indicator of improper upbringing<br>• Romantic love also caused embarrassment |
| Self- ignorance about sex | Passive learning<br>Very little input about sex<br>Informal learning | • From own experience<br>• Almost zero input from family members<br>• Informal learning mainly from religious classes, focus more on rights and responsibilities, permissible (or nonpermissible) techniques for sex. |
| Lack of husband's role in mutual sexual enjoyment | Bad temper<br>Not emulating the Prophet in his daily treatment of his wife | • Bad temper was a common attitude in some husbands<br>• Unable to understand or respond to their wife's sexual needs<br>• Lack of foreplay and post-play<br>• Lack of ongoing support in marriage |

3.2.1. Sex Is Taboo and Culturally Unacceptable

Taboos about sex were well-known in this group of Malay women. They considered it disgraceful to talk about sex. Many of them had learned little about sexuality from their families. As shared by Suzila, a 29-year-old social worker who experienced difficulties with sexual desire and having orgasms, "Sex was not talked about in the family because it was seen as a shameful thing."

Characteristic with a Malay sense of reticence, the women euphemistically referred to sex as "benda dalam kelambu" ("the thing inside the mosquito net"). Sex talk was deemed personal and a confidential matter between husband and wife, not to be openly discussed with others or those unmarried.

> Malay society is quite discreet. When anyone talks about sex, everyone feels that is the one thing that no one needs to discuss, it relates to couple relationships. Perhaps [it is] because of our culture, religion is not the problem. (Haslinda, 41, lecturer, difficulties with orgasms and sexual desire)

The women saw avoidance of sex-related words as an indicator of a proper upbringing, that is, one accorded with cultural and religious teachings.

> Malay people do not talk about sexual matters. Mother and father would be an-gry, because that thing is a secret of married couples. If we unintentionally speak about it, they will say we are talking about an obscene thing, or whatever. [We] cannot talk about that. (Bibi, 50, nanny, difficulties with orgasms)

A few women still perceived sexual intercourse itself as something dirty, even though they believed it was lawful after marriage. For example, from her reading, Aisyah perceived sexuality as something disgusting. She felt sex was not really important compared with the other family roles that she was expected to perform as part of Malay culture.

> I don't feel sexuality is a great necessity. It is just more towards responsibility. I do not feel that [sex] is a priority. When I read, I feel like that thing is too disgusting . . . [I feel weird] about the extent to which they [other people] position the sex in their life. (Aisyah, 42 years, scientist, orgasm and sexual desire problems)

Many women also saw pornography as sickening and not culturally or religiously acceptable. Julia described her childhood experience, of accidentally seeing oral sex photos, as a "disgusting experience" that stuck in her mind and led her to dislike sex,

> I had never seen such pictures; only a small book too. What was that man doing with that woman? That woman was licking that man's penis in that picture. After that, I felt disgusted to see it. I felt afraid. "Why would grown-up people do that?" That event was still flashing back in my mind till now, although it occurred during my childhood. (Julia, 40 years, government officer, lack of sexual desire problem)

Practising oral sex was perceived to be unacceptable even among married couples since licking the genital area was judged "unclean." From their Islamic teachers, they understood the meaning of sex as penile–vaginal intercourse only. Oral sex was seen as something that was new to some women, and also not talked about openly in Malaysian society. Unlike anal sex, it is not totally prohibited in Islam.

Romantic love was another behaviour that women believed was not practiced widely in Malay society, even within the privacy of the home. It was considered peculiar for people to express love this way publicly.

> Kissing . . . not really, for Malay people. When my husband started his work, we stayed with my mother. Every time my husband went out for work, we shake hands and he kisses my brow. Then many neighbours started to ask, "How about today, you already kiss his hands? He kisses your brow?" Meaning . . . those things were so weird. (Haslinda, 41 years, lecturer, sexual desire and orgasmic problems)

Demonstrating romantic love in front of their children was seen to conflict with the desire to take good care of their children. The attributes of shyness, encouraged by Adat, meant that open displays of affection caused embarrassment. No public displays of affection, within or outside the home, were considered. Alya explained it this way,

> We want to kiss in front of our children, but we are shy. We may want to praise a wife in front of mum and dad, or want to feed a wife with his own hand when eating together, or want to give a surprise to a wife, or to make a husband happy but this was not considered important. Other things are more important . . . taking care of children is of more concern . . . not keeping the spark in the husband and wife relationship. (38 years, teacher, orgasm problems)

However, for several women, couples tickling or massaging each other did occur and were viewed as signs of intimacy and love. Others showed their love and respect to family members by shaking hands, kissing their husband's hands or being kissed by their husband on the forehead, especially during Hari Raya (Eid festivals), which is the most auspicious annual festival for Muslims in Malaysia (and other parts of the world).

> Every night my father asked my mother to picit-urut [massage]. I have never seen our parents showed romantic actions in front of us, like hugging. Perhaps they were afraid what their children will say or feel embarrassed. When my big brother with his wife were tickling each other, in a private area and I accidentally saw them [doing that], they quickly stopped. (Husna, 27 years, clerk, loss of sexual desire)

When listing taboos, a few women remarked that their beliefs about such things were changing due to developing a better understanding of their rights and Islamic teachings. "People say sex is dirty but not me. Only in Malay Adat. Not Islamic teaching. Sex is also for us" (Haslinda, 41 years, lecturer, sexual desire and orgasm problems).

### 3.2.2. Self-Ignorance about Sex

Apart from taboos, traditional Malay society also views sexuality as a natural function and therefore not necessary to be taught by families or communities, either directly or indirectly. Amnah's experience of learning about sex only after her marriage is typical. Amnah's husband was her "so-called" first source of information about sex. She said: "After I was married, by experience I know; what [an] intimate relationship is, and feeling in love." (40, government servant, loss of sexual desire)

Haslinda also shared further insights into societal perceptions on this matter.

> When [the authorities] in Malaysia wanted to implement sex education, elderly people said "Ahhh! Even blind people can [know what to do]." It means that there is no need to learn. Like eating rice, they said, "When hungry, this hand moves, this mouth automatically opens widely. Even without proper [sex] education, they're still able to have many children. (41, lecturer, difficulties with sexual desire and orgasms)

For some, even the act of searching for sex-related information in books or through other people to gain new knowledge made them feel ashamed.

> I've never been taught about this. I just read it on the internet. If from books . . . just a few, my friends gave during our marriage, but I never buy them. I want to buy but feel embarrassed. But I did buy presents for my friends, books like "Marriage in Islam" and "The Stories of the Prophet's Household". These topics do not really look as though they are focusing on sexuality. (Murni, 32, lecturer, lack of sexual desire)

However, most of the women with sexual difficulties also realised that ignorance about sexual relationships, sexual functions, and responses was among the factors contributing to their inability to have better sexual relations. Alya said the lack of social support to learn sexual skills before marriage might be one reason for what she had to go through.

> "You are like angel," he said. Why? Because when we did that, I did not know how, I was very stiff, not aroused, because early in the marriage I felt weird, I had never been exposed to sexual [things] before. I started to think, oh, sex is like this. Starting from then I started to learn, what a man wants. (38, teacher, difficulties with orgasms)

Zaini, a 33-year-old teacher who suffered from vaginismus (pain during sex), had misconceptions about sex early in her marriage: "I did not know that after marriage [I] would have to have sexual intercourse. At that time, I was very surprised. The thought of having someone touching me like that."

This response was similar to the women who had difficulties with orgasms. The majority of these women only realised that they had a problem after hearing about orgasms from other people. For these women and their partners, lack of knowledge about sex, including stimulation, meant their inability to feel pleasure from sexual intercourse was a common occurrence.

> I do not know the exact time when it [sexual difficulties] started to happen because I did not know what sex was all about. So when I did not know whether I experienced a 'normal' sexual encounter [climax] or not, I was not sure. When chatting with my friends, I said: "Oh, is it like that?" (Aisyah, 42, scientist, difficulties with sexual desire and orgasms)

Due to their ignorance, most participants felt that books were among the best ways to gain early exposure and knowledge about sex. Maryam remarked: "I read books. It teaches me the way to have sex" (51, housewife, loss of sexual desire). Realising the impact of the lack of sexual knowledge, many participants wished that healthcare professionals and Islamic scholars had provided them with proper sex education. With hindsight, they felt that they could have avoided the challenges they experienced.

> I want an easy-going female doctor, able to understand our problem; there are many opinions on the outside and inside, about intimacy matters, the ways to be together, ways to have happiness, and some ways that we can and can't do it in line with our religious beliefs. (Sofiah, 30, housewife, loss of sexual desire)

### 3.2.3. Lack of Husband's Role in Mutual Sexual Enjoyment

While the women interviewed in this study believed that their sexual challenges were multifactorial, many reported constant emotional frustration with their husband's attitude and personality, which they felt were not following what Islam and Adat instruct. This manifested in a lack of foreplay and post-play during the sexual relationship and/or temper tantrums, or a lack of on-going support in their marital relationship. They felt their husbands were not meeting their responsibilities in the marriage, and this was an important contributor to how they felt about their sexual experiences.

Some husbands seemed unable to understand or to respond to their wife's sexual needs during the sexual relationship. Alya spoke of how her husband treated her roughly during sex and with an absence of adequate foreplay. She was emotionally and physically hurt to the extent that her sexual desire dissipated and she was unable to achieve orgasm.

> I think my husband is rough, meaning he does [intercourse] in a rough-mannered way. If he touches me, I feel pain. He is so rough; it feels like tearing my private part. I just hug him, and then he goes in straight away. He does not really want to touch or rub my body, perhaps give me some kisses. No. (38, teacher, difficulties with orgasms)

These experiences were similar to Ziela and Mona. They claimed that their husbands' inability to find out and do what the wife wanted during intercourse affected their sexual pleasure.

> As he stimulates, or the way he wants to kiss . . . there are ways that I don't like. I have to do it myself. Man a little bit greedy [laughing], I want something slow. (Ziela, 35, pharmacist, difficulties with sexual desire and orgasms)

> He does foreplay, but does not berlaga angin [meet expectations], I don't like it [oral sex]. I feel angry. (Mona, 29, housewife, loss of sexual desire)

Bad tempers were a common emotional burden that led to the loss of sexual desire and even though discouraged in Muslim marital relationships, still occurred among a few couples. For example, Mazni expressed her feelings of stress and showed how she accommodated her husband's nasty temper through suppressing her own sexual desire.

> My husband is a bit aggressive. If he is angry with something else, I would also feel like he is angry at me. If he is a little bit rough with our children, I can't

accept it. At last, there brings hatred in my heart, for his actions. (51, housewife, loss of sexual desire)

Similarly, for Roza, who could not tolerate her husband's temper. It stopped her from wanting sex, even though her husband helped with household chores at other times.

My husband is really good, he manages it [household chores] all. But when he hears the baby crying, especially when he is sleeping, he turns bad-tempered, a 360-degree change from jolly to grumpy. The love in my heart shatters when he is angry. (42, teacher, lack of sexual desire)

Breach of trust was reported as another cause of dissatisfaction with husbands. Suzila, who had never experienced an orgasm, attributed her diminished desire to rapid or hasty sex, no open discussion, and feelings of betrayal.

He . . . [is always] in a hurry. A long time but I've never got it; he quickly finishes, well better no need [to do]. I told him," I want like this and like this," if he can do it . . . but men are difficult. I also feel disgusted with him. He has committed adultery before. (29, social worker, sexual desire and orgasm problems)

Alya, who was in a polygamous marriage, added that she felt her husband was ignorant of his marital responsibilities and that his different treatment of his wives was unfair.

For him, that thing is not important; his job is more important. No tutelage and affection in the family. After he married his second wife, he often calls me black, kind of insulting . . . before this, he never said so. With his young wife, he can say, "I love you," "I miss you." Can hold her hands. But not with me, even after I asked for it. If he walks with me, I'm walking here; he's walking there. It seems like he does not want other people to know, that's his wife. My heart really hurts. (38, teacher, difficulties with orgasms)

Unequal gender relations in marriage gave rise to feelings of resentment, particularly women with husbands from the east coast of Peninsular Malaysia, where marriage roles are more Adat-influenced. Despite believing in the importance of unity and the mutuality of roles and responsibilities of husbands and wives, these women expressed dissatisfaction with their assigned roles. They explained that tradition and belief systems in these places expected women and wives to play larger roles in family matters than men. For example, besides working, women had to care for their husbands and children; men only had to support the family. At home men would rest and not have to deal with domestic work. It is still very much a patriarchal society.

He was the youngest child, what he wants, his mother and sisters are there, people from the east coast. I have to do everything; I become phobic . . . [He] is sort of lazy. He is a gregarious type, and [spends] less time with our children. So many things I have to do, but he just wants that one thing [intercourse]. (Haryati, 40, lecturer, lack of sexual desire)

Ziela, also from the east coast of Peninsular Malaysia had a similar problem, although she was married to a man from the west coast. Unlike other women who blamed the patriarchal Adat system, she believed that among the causes of her problems was her husband's failing to emulate the Prophet's treatment of his wife.

Actually, in Islam [it] is not like that, look at what the Prophet did . . . the best to his wives, assisting them in everything . . . not like him [her husband] . . . people called, be the boss. Just sitting, waiting for food and drinks to be served. (35, pharmacist, difficulties with sexual desire and orgasms)

Some women recognised the frequency of sex demanded by their husbands as a factor that affected their sexual functioning. For others, it was not just a matter of frequency but the quality of sex, especially at night, when they felt tired. Izura was one woman who identified the incompatible sexual desires in her relationship, even though she believed

she had to provide him with sex as this was one of her marital roles. She did not reject this role but also believed that within marriage, there should be mutual understanding on this matter.

> My husband sort of has strong sexual desires. Till I said, "Abang [husband], I have to find medicine." Every day, he wants it, I'm so tired. Tired because I do not [have the same desire] like him. (34, clerk, lack of sexual desire)

Ziela too felt that nightly sex was not conducive to [her sexual pleasure].

> Perhaps the time is not suitable, I'm very sleepy, though he kisses or does something, I'm not excited. Too lazy to reply. Not motivated. Not aroused. (35, pharmacist, difficulties with sexual desire and orgasms)

A husband's failure to respect his wife's social obligations affected the latter's emotions and overall sexual functioning as well. For some women, keeping promises made to their parents was an important aspect of their religious obligations, which for them, resulted in a trusting and blissful marriage. Problems arose when husbands ignored such promises or did not make time for their wives to meet these commitments. For example, Mona traced her disappointment to her husband not appreciating her familial bonds or not allowing her to stay with her mother.

> During confinement [I] have stayed at mother-in-law's home. He promised to send me back to my mother's house after 40 days. I felt crazy because I missed my mother because she is sick. I cried and felt frustrated because he did not do what he promised. (29, housewife, loss of sexual desire)

All the women interviewed identified their husbands' knowledge deficit or lack of ability in applying knowledge as a major challenge, which in turn had a negative impact on their sexual experiences. These areas of knowledge covered issues both internal to the relationship (e.g., women's emotional and sexual needs) and external to it (e.g., marital responsibilities and social obligations).

## 4. Discussion

The women interviewed saw the notion of sex as something "dirty" prevailing among Malays—as a subject that should not be discussed, let alone learnt prior to, or even after marriage. Cultural and religious misconceptions appeared to have a negative influence on their sexuality. As a phenomenon that forever resurfaces, religion is currently discussed through the lens of the sociocultural aspect and sexual issues (Öztürk 2019). The findings were at par with those found by Kuru (2014), who objected to arguments about Islamic patriarchy resulting from Islam's innate authoritarianism, and rather saw these issues as due to the influence of sociocultural (Adat) factors within countries such as Malaysia.

Women in this study highlighted that sexual behaviours—such as expressing love or displaying any romantic gestures in the form of holding hands or intimate touching in public—should be discouraged. However, the women did want emotional intimacy and to be shown love and respect for how they felt and what they desired. Instead, as seen in the narrative above, some reported that their husbands lacked sexual knowledge and maturity.

Hoesni et al. (2013) found comparable findings; Malays displayed acquiescence to the societal norm of expressing love through "sharing and supporting communication", rather than "commitment-faith" or "romantic-physical". "Commitment-faith" refers to the ideal way for a Muslim couple to express love, for example, by ensuring a spouse's safety and maintaining "acceptable" sexual acts between them. Least practiced among Malays were "romantic-physical" expressions of love, such as holding hands, caressing, articulating loving words, or giving sexual massages or bathing together, believed to be popular among those in the West.

In fact, expressing romantic love and affection between a husband and wife/wives is highly recommended in Islam. The Prophet Muhammad said: "The best of you is the best to his wives, and I am the best of you to my wives" (cited in At-Tirmidhi 2007, pp. 530–31).

Some hadith strongly encourage a husband to take good care and show affection to his wife in the best way that he can, and even to "play" with or caress her. Aisya, one of the Prophet's wives, shared how romantic the Prophet was with her:

> "Yes. The Messenger of Allah would call me to eat with him while I was menstruating. He would take a piece of bone on which some bits of meat were left and insist that I take it first, so I would nibble a little from it, then put it down. Then he would take it and nibble from it, and he would put his mouth where mine had been on the bone. Then he would ask for a drink and insist that I take it first before he drank from it. So, I would take it and drink from it, then put it down, then he would take it and drink from it, putting his mouth where mine had been on the cup." (cited in An-Nasai 2007, pp. 224–25)

Many women in this study wanted to see a shift towards an acceptance of public behaviours permitted in Islam, such as kissing on the forehead and holding hands. Others still preferred to adhere to Adat traditions.

Self-ignorance about sex was also associated with a higher chance of experiencing sexual difficulties. The women believed that they themselves lacked the ability to resolve this problem—improving their sexual knowledge and skills—due to prevailing taboos and lack of support from their husbands. Elsewhere too, Halvorsen and Metz (1992) and Muhamad et al. (2016) have highlighted self-ignorance about sex as one of the aetiologies for FSD.

Empirical evidence suggests sexual educational intervention is an effective way to help lower hypoactive sexual desire disorder (Kaviani et al. 2014). In Malaysia, premarital courses have been introduced to ensure those planning on getting married first understand their roles and responsibilities within marriage, with additional sessions from the medical and Islamic perspectives of sexuality.

Like other women, including those of different faiths in predominantly Muslim or Christian nations (see Muhamad et al. 2016; Vangelisti and Alexander 2002), the Malay women in this study cognitively struggled to accommodate the roles of a "good wife" and "good mother" with their own needs. These findings are consistent with Mitchell et al. (2011), who found that British women negotiated their meanings of sex and their roles as wives according to their religion, and that those unable to resolve sexual and relationship problems adjusted their expectations and fitted themselves to their circumstances. Despite sexual difficulties, they would still engage in sex, but with expectations lowered to having just "good-enough" sex. These British women also sought to normalise their sexual problems and/or chose strategies for avoiding sex.

Notwithstanding these similarities, the reasons for negotiation among the British women in Mitchell's study (2011) and the Malay women of this study were different. The latter dropped their expectations and agreed to sexual intimacy as a service to God when perceived as the primary responsibility of a Malay wife in acknowledging her husband's rights (Mohd 2009). A religious meaning of sexual activity and acceptance of sex without pleasure was also found by Hoel and Shaikh (2013) in their study of the sexuality of South African Muslim women.

As noted above, a number of women attributed the cause of their FSD to their husbands. Several referred to the "traditional" upbringing of Malay men, whereby they do little to maintain a household. Such behaviour goes against what Islam promotes as the responsibilities of a man in marriage. Specifically, they are expected to guard, protect, and provide fair treatment to wives (al-Qardawi 1960). This role is highlighted in the following hadith,

> "Aisha, the wife of the Prophet Muhammad (pbuh), was asked, "What did the Prophet use to do in his house?" She replied, "He used to keep himself busy serving his family and when it was the time for prayer, he would go for it.". (cited in Al-Bukhari 1997, p. 385)

In the Qur'an, men and women are mentioned as "garments" fitting each other: "They are as a garment for you, and you are a garment for them" (Al-Quran 2013, 2:187). Sexual duality in creation, which enables them to unite, is recognised by Muslims as one of the great signs Allah has bestowed on humankind: "And among His signs is this: He creates for you mates out of your own kind, so that you might incline towards them and He engenders mutual love and compassion between you" (Al-Quran 2013, 30:21). "He has created you from a single soul and from that soul He created its mate" (Al-Quran 2013, 4:1). The sexual act through marriage is aimed to "receive spiritual strength of the unity of two souls" (Ashraf 1998). Sexual relationships are viewed as a means of bringing a couple to worship God. Failure by Malay men to fulfil their religious obligations as husbands is another factor causing FSD among women (Pan American Health Organization (PAHO) and World Health Organization (WHO) 2000).

Other interpersonal factors, such as a husband's intolerable behaviour or incompatible sexual needs, were also identified as challenges. Women had difficulty accepting when their husbands ignored their emotions, were short-tempered and aggressive, failed to help with housework or with nurturing their children, and when they fell short in fulfilling their sexual needs with more consideration. They wanted emotional intimacy and to be shown love and respect for their moods and desires. Some also reported that their husbands lacked sexual knowledge and behaved immaturely.

These issues created an inability to adapt to differences in interpersonal scripts and caused women to reject their husbands' behaviours both cognitively and physically, which manifested in their sexual "dysfunction" and lack of fulfilment from sex. This pathway described by McCabe (1991) is consistent with those of other Asian studies that have highlighted significant associations between FSD and distorted or inadequate marital knowledge of sexual dissatisfaction, especially the absence of foreplay in sex (Lo and Kok 2014; Tehrani et al. 2014).

These experiences are not just found within Malay, Asian, or Muslim populations, as similar findings have been reported in qualitative studies conducted in western countries, where poor sexual techniques and partners' behaviour during sex have been shown to have severe adverse effects on women's sexual functioning (Bellamy et al. 2013; Murray et al. 2014; Sutherland 2012).

## 5. Conclusions

This qualitative study provides us with a better understanding of the phenomenon of FSD among Malay women. Those in this study described their perceived causes, concerns, and expectations about their problems and relationships in ways that appear to conform to a particular Malay-Muslim narrative that underscores the cultural expectations of an ideal "never-to-be-questioned" Malay husband and a good, obedient Malay wife. The implications of these findings can potentially help healthcare providers and Islamic scholars to find better remedies for unsatisfactory sex among married Malay couples. The personal accounts from this study have implications for sexual therapy where the therapist could focus more on romance and eroticism in sexual acts. The expression of love and sexual satisfaction is pivotal, and needs to be emphasized, especially during counselling and consultation in primary care and specialist clinics.

The male and female sexual relationship in marriage is a sacred duty as stated in the Quran, as this relates to how humans are connected to God's creative power. Many hadith also explain how Muslim married couples should behave and perform their roles and responsibilities. Not adhering to this calling and a lack of understanding of the source of sexual pleasure are major causes of FSD. Healthcare sexual providers and Islamic scholars can play an important role in helping to overcome some taboos and ill-informed knowledge associated with sex, which should be reinforced even earlier in sexual education in the school-based programs (Khalaf et al. 2014). Unawareness in the community and lack of active involvement in sexuality education are key areas of concern. These findings

should contribute towards a more enriched curriculum in formal sexual education classes and courses.

**Author Contributions:** Conceptualization, R.M. and P.L.; methodology, R.M. and P.L.; software, R.M.; validation, R.M., D.H., W.Y.L., H.S. and P.L.; formal analysis, R.M., D.H., W.Y.L. and P.L.; writing—original draft preparation, R.M. and M.M.Z.; writing—review and editing, D.H., W.Y.L., H.S. and P.L. All authors have read and agreed to the published version of the manuscript.

**Funding:** This research received no external funding.

**Institutional Review Board Statement:** The study was conducted according to the guidelines of the Declaration of Helsinki, and approved by the La Trobe University Ethics Committee (protocol code of HEC13-025 15th August 2013); Research Promotion and Co-ordination Committee Economic Planning Unit, Prime Minister's Department of Malaysia (protocol code of UPE: 40/200/19/2979 on 8th April 2013); Medical Ethics Committee of Universiti Malaya Medical Centres (protocol code of 996.4 on 4th June 2013); Human Research Ethics Committee USM (protocol code 00007718; IRB registration number: 0004494 on 27th August 2013) and UKM Research Ethics Committee (protocol code of 1.5.3.5/244/FF-2014-086/Clinical Professor Dr Hatta Sidi on 22nd February 2014).

**Informed Consent Statement:** Informed consent was obtained from all subjects involved in the study.

**Acknowledgments:** Special thanks to all participants, staffs and friends who helped us in completing this study in La Trobe University, Universiti Sains Malaysia, Universiti Kebangsaan Malaysia, and Universiti Malaya.

**Conflicts of Interest:** The authors declare no conflict of interest.

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
