# Peer review of "Transcripts of Unfulfillment: A Study of Sexual Dysfunction and Dissatisfaction among Malay-Muslim Women in Malaysia"

_religions, doi:10.3390/rel12030205_

Round 1
Reviewer 1 Report
Overall, I found this to be a well-designed study that reported on an under-reported area of social life that may be relevant in other Muslim communities. The methodology is sound, as is the contextualizing literature. More literature could be added on sexual education with Muslim youth and young adults; for example, the work of Fida Sanjakdar. Definitely, describe how the interviews were translated into English, and when. Please also check that the lines match up on the table. Finally, the paper could use copy editing. Wishing you all the best!
Author Response
Dear reviewer,
Thank you for your comments and suggestions.
|
Comment |
Response |
1 |
Reviewer 1 Overall, I found this to be a well-designed study that reported on an under-reported area of social life that may be relevant in other Muslim communities. The methodology is sound, as is the contextualizing literature. Wishing you all the best!
|
Thank you. |
|
More literature could be added on sexual education with Muslim youth and young adults; for example, the work of Fida Sanjakdar. |
Thanks for the recommendation. Unfortunately, we can’t download the Fida Sanjakdar work freely, so we use research work from our country.
In the introduction: page 1-2 For Malay women, education of their sexual identity starts in girlhood through their family upbringing and local religious teachers and it continues through marriage and motherhood (Omar 1994). Apart from politeness and humility, loyalty is also seen as a trademark attribute of the Malay identity (Goddard 2000). Nevertheless, beyond these cultural attributes, limited knowledge of sex or ways to address deficiencies in sexual relationships, coupled with non-participation of husbands in resolving these issues seem to underlie the problems experienced by Muslim women (Khoei et al. 2008; Muhamad et al. 2016). Government programs have been in place in Malaysia since 1996, that provide avenues to address these issues. Adolescent health services were started by the Ministry of Health in 1996, mainly within school health units but these were not fully utilized until 2018 when the Ministry rebranded and strengthened them as adolescent-friendly health services in health clinics (Awang et al. 2020; Nik Farid et al. 2018). Since 2011, trained teachers have delivered the Health Reproduction and Social Education (PEERS) program in schools during Islamic Studies, Moral Studies, and Science and Health Education classes (Awang et al. 2020; Nik Farid et al. 2018). However, PEERS is not a comprehensive sexual health program and is still considered controversial among many parties (Mohd Mutalip and Mohamed 2012). The compulsory premarital courses, also introduced in 1996 by the Department of Islamic Development Malaysia, using Integrated Module for Premarital Course, mainly covers marital roles and rights (Saidon et al. 2016).
In the discussion: page 11-12 we add: In the Qur’an, men and women are mentioned as ‘garments’ fitting each other: They are as a garment for you, and you are a garment for them (2: 187). Sexual duality in creation which enables them to unite is recognised by Muslims as one of the great signs Allah has bestowed on humankind: And among His signs is this: He creates for you mates out of your own kind, so that you might incline towards them and He engenders mutual love and compassion between you (the Qur’an 30:21). He has created you from a single soul and from that soul He created its mate (The Qur’an, 4:1). The sexual act through marriage is aimed to “receive spiritual strength of the unity of two souls” (Ashraf 1998). Sexual relationships are viewed as a means of bringing a couple to worship God. Failure by Malay men to fulfil their religious obligations as husbands is another factor causing FSD among women (PAHO and WHO 2000).
In the conclusion: Page 12 Healthcare sexual providers and Islamic scholars can play an important role in helping to overcome some of the taboos and ill-informed knowledge associated with sex and it should be reinforced even earlier in the sexual education school-based program (Khalaf et al. 2014).
References added:
Al-Quran. 2:187, 2:222-223, 4:1. 2013. Translated by Abdullah Yusuf Ali. Ware, Hertfordshire: Wordsworth Editions Limited. Ashraf, Syed. (1998). The concept of sex in Islam and sex education, Muslim Education Quarterly 15(2):37-43. Awang, Hafizuddin, Azriani Ab Rahman, Surianti Sukeri, Noran Hashim and Nik Rubiah Nik Abdul Rashid. 2020. Adolescent-friendly health services in primary healthcare facilities in Malaysia and its correlation with adolescent satisfaction level, International Journal of Adolescence and Youth 25(1): 551-561, doi: 10.1080/02673843.2019.1685556. Khalaf Zahra Fazli, Wah Yun Low, Effat Merghati-Khoei, and Behzad Ghorbani. Sexuality education in Malaysia: perceived issues and barriers by professionals. Asia Pacific Journal of Public Health 26(4):358-366. doi: 10.1177/1010539513517258. Saidon, Rafeah, Amal Hayati Ishak, Baterah Alias, Fadhilah Adibah Ismail, Suliah Mohd Aris. 2016. Towards Good Governance of Premarital Course for Muslims in Malaysia. International Review of Management and Marketing 6(S8): 8-12. Available online: http: www.econjournals.com.
|
|
Definitely, describe how the interviews were translated into English, and when. |
Only the quotes that has been chosen to be used in the thesis were translated in English. Page 3 lines 113-114 |
|
Please also check that the lines match up on the table. |
Line in Table 1 has been amended. We also add visible lines in the Table 2. |
|
Finally, the paper could use copy editing. |
Thanks. One of the authors is a native English speaker and we have thoroughly checked this manuscript. |
The changes were highlighted in red in the manuscript.
We look forward to hearing from you. We would be glad to respond if any further comments that you may have.
Reviewer 2 Report
I read the article namely Transcripts of Unfulfillment: A Study of Sexual Dysfunctionality and Dissatisfaction Among Malay-Muslim Women in Malaysia. I think this is a very good article and needs only a little bit modification regarding the role of Islam in politics in society in general. In this regard, the article should use these studies and add only one very small part on how and why Islam is an important element for general socio political issues. The author should use these resources:
Öztürk, A. E. (2019). An alternative reading of religion and authoritarianism: the new logic between religion and state in the AKP’s New Turkey. Southeast European and Black Sea Studies, 19(1), 79-98.
Kuru, Ahmet T. Islam, authoritarianism, and underdevelopment: A global and historical comparison. Cambridge University Press, 2019.
Ozturk, Ahmet Erdi. Religion, Identity and Power: Turkey and the Balkans in the Twenty First Centruy. Edinburgh University Press, 2021
Author Response
2 |
Reviewer 2 I read the article namely Transcripts of Unfulfillment: A Study of Sexual Dysfunctionality and Dissatisfaction Among Malay-Muslim Women in Malaysia. I think this is a very good article and needs only a little bit modification regarding the role of Islam in politics in society in general. In this regard, the article should use these studies and add only one very small part on how and why Islam is an important element for general socio political issues.
The author should use these resources:
Öztürk, A. E. (2019). An alternative reading of religion and authoritarianism: the new logic between religion and state in the AKP’s New Turkey. Southeast European and Black Sea Studies, 19(1), 79-98.
Kuru, Ahmet T. Islam, authoritarianism, and underdevelopment: A global and historical comparison. Cambridge University Press, 2019.
Ozturk, Ahmet Erdi. Religion, Identity and Power: Turkey and the Balkans in the Twenty First Centruy. Edinburgh University Press, 2021 |
Thank you for comments and recommendations. We have added the role of Islam in Malaysian politics in society in general and how politics influence sexuality understanding in the page 1 using evidences from Malaysia.
Introduction section:
Malays have a unique legal identity formed by their religion (Islam), language (Bahasa Melayu), and customs (Adat) formalized in Malaysia’s independence in 1957 (Federal Constitution 2010). These three components were brought together as a political solution in a geographical area with a complex history but also contribute to shaping the sexual identity of Malay women. While Islam steers Malaysia through its beliefs, religious practices and political responses, Adat underpins how people live and language holds everything together (Mastor et al. 2000). The process of integration, particularly of Islam and Adat, presents opportunities to influence change. Despite its position as the official religion of Malaysia, Islam has not totally overruled Malay Adat. While mostly Adat and Islam work together, some Adat practices are non-sharia, that is, they do not comply with Islam (Harun 2009). However, over time non-sharia practices are being slowly modified to follow Islamic law both through political will and through the influence of Islam on what constitutes desirable personal attributes (Harun 2009). For example, Malays are associated with politeness and humility, which are viewed as positive characteristics and are actively reinforced by Islamic teaching. Government members and Islam scholars have initiated various ongoing campaigns to encourage values consistent with Islamic teaching to shape a respectful Malay society with high morality and a strong identity (Ahmad and Mohamad Nasir 2018; Malaysiakini 2018).
While in the discussion section: Page 10 The women interviewed saw the notion of sex as something ‘dirty’ prevailing among Malays; as a subject that should not be discussed, let alone learnt prior to, or even after marriage. Cultural and religious misconceptions appeared to have negative influence on their sexuality. As a phenomenon that forever resurfaces, religion is currently discussed through the lens of the sociocultural aspect, and sexual issues. (Öztürk 2019). The findings were at par with those found by Kuru (2014) that objects to arguments about Islamic patriarchy resulting from Islam’s innate authoritarianism, and rather sees these issues as due to the influence of socio-cultural (Adat) factors within countries such as Malaysia.
Ahmad, Jamilah, and Nur Nasliza Arina Mohamad Nasir. 2018. Malaysian Journal of Youth Studies 19:168-198. Available on: https://iyres.gov.my/images/MJYS/2018/MJYS%20Vol%2019%20Dec%202018-172-203.pdf (accessed on 13 March 2021). Harun, Yaacob. (2009). Islam and Malay culture [Blog post]. Available online: https://yaacob.wordpress.com/2009/05/24/articles-on-culture/ (Accessed on 1 August 2016) Kuru, Ahmed T. 2014. Authoritarianism and democracy in Muslim countries: rentier states and regional diffusion. Political Science Quarterly, 129(3): 399-427. Available online: https://politicalscience.sdsu.edu/docs/Kuru/Kuru_PSQ_Muslim.pdf Malaysiakini. 2018. Melayu perlu ubah sikap, sistem nilai - Dr M. Available online: https://www.malaysiakini.com/news/458050 (Accessed on 121 March 2021) Mastor, Khairul Anwar, Jin Putai, and Cooper, Martin. 2000. Malay culture and personality: A big five perspectives. The American Behavioural Scientist 44(1): 95-111. doi:10.1177/00027640021956116 Nik Farid, Nik Daliana, Mohd Faris Mohd Arshad, Nur Asyikin Yakub, Rafdzah Ahmad Zaki, Haslina Muhamad, Norlaili Abdul Aziz, and Maznah Dahlui. 2018. Improving Malaysian adolescent sexual and reproductive health: An Internet-based health promotion programme as a potential intervention. Health Education Journal 77(7):837-848. doi:10.1177/0017896918778071 Öztürk, Ahmet Erdi. 2019. An alternative reading of religion and authoritarianism: the new logic between religion and state in the AKP’s New Turkey. Southeast European and Black Sea Studies 19(1):79-98.
|
The changes were highlighted in red in the manuscript.
We look forward to hearing from you. We would be glad to respond if any further comments that you may have.